# Deep Learning Multi-Domain Model Provides Accurate Detection and Grading of Mucosal Ulcers in Different Capsule Endoscopy Types

**DOI:** 10.3390/diagnostics12102490

**Published:** 2022-10-14

**Authors:** Tom Kratter, Noam Shapira, Yarden Lev, Or Mauda, Yehonatan Moshkovitz, Roni Shitrit, Shani Konyo, Offir Ukashi, Lior Dar, Oranit Shlomi, Ahmad Albshesh, Shelly Soffer, Eyal Klang, Shomron Ben Horin, Rami Eliakim, Uri Kopylov, Reuma Margalit Yehuda

**Affiliations:** 1Penta-AI, Tel Aviv 6701101, Israel; 2Faculty of Medicine, Ben-Gurion University of the Negev, Be’er Sheva 8410501, Israel; 3Department of Internal Medicine E, Sheba Medical Center, Tel Hashomer, Ramat Gan 5262100, Israel; 4Sackler School of Medicine, Tel Aviv University, P.O.B 39040, Tel Aviv 6997801, Israel; 5Department of Gastroenterology, Sheba Medical Center, Tel Hashomer, Ramat Gan 5262100, Israel; 6Department of Internal Medicine F, Sheba Medical Center, Tel Hashomer, Ramat Gan 5262100, Israel; 7Internal Medicine B, Assuta Medical Center, Ashdod, Israel, Ben-Gurion University of the Negev, Be’er Sheva 8410501, Israel; 8Sami Sagol AI Hub, ARC, Sheba Medical Center, Tel Hashomer, Ramat Gan 5262100, Israel

**Keywords:** machine learning, capsule endoscopy, Crohn’s disease

## Abstract

Background and Aims: The aim of our study was to create an accurate patient-level combined algorithm for the identification of ulcers on CE images from two different capsules. Methods: We retrospectively collected CE images from PillCam-SB3′s capsule and PillCam-Crohn’s capsule. ML algorithms were trained to classify small bowel CE images into either normal or ulcerated mucosa: a separate model for each capsule type, a cross-domain model (training the model on one capsule type and testing on the other), and a combined model. Results: The dataset included 33,100 CE images: 20,621 PillCam-SB3 images and 12,479 PillCam-Crohn’s images, of which 3582 were colonic images. There were 15,684 normal mucosa images and 17,416 ulcerated mucosa images. While the separate model for each capsule type achieved excellent accuracy (average AUC 0.95 and 0.98, respectively), the cross-domain model achieved a wide range of accuracies (0.569–0.88) with an AUC of 0.93. The combined model achieved the best results with an average AUC of 0.99 and average mean patient accuracy of 0.974. Conclusions: A combined model for two different capsules provided high and consistent diagnostic accuracy. Creating a holistic AI model for automated capsule reading is an essential part of the refinement required in ML models on the way to adapting them to clinical practice.

## 1. Introduction

Capsule Endoscopy (CE), in clinical use since 2000, is a reliable and noninvasive diagnostic tool that revolutionized the assessment of small-bowel mucosa [1,2,3,4,5]. CE is a sensitive and accurate clinical tool for diagnosing and monitoring Crohn’s disease (CD) [3,4,6,7,8,9,10] and has good prognostic value for relapse in clinical remission [11]. CE is recommended in international Crohn’s disease guidelines together with cross-sectional imaging [12,13]. Despite the well-described merits of CE, the clinical performance of this modality may be further augmented by shortening reading time, improving interobserver variability, and implementing precise scoring algorithms. In the past few years, artificial intelligence (AI) deep learning algorithms, termed convolutional neural networks (CNN), have revolutionized the computer vision field, offering remarkable near-human accuracy in different image analysis tasks, including medical image analysis [14]. Several research groups have tested the ability of AI algorithms to diagnose various lesions in the small intestine by CE, including bleeding [15,16,17], angioectasia [18,19,20], intestinal stricture [21], celiac disease signs [22,23] and hookworm infection [24], achieving high sensitivity and specificity. In CD, deep learning has been proven accurate in detecting and grading ulcers and strictures on CE [21,25,26,27,28,29,30,31,32,33]. There is still a long way to go before the implementation of AI-based capsule reading algorithms in clinical practice, and there are several challenges. The main potential obstacles for patient-level implementation are the marked variability of images between examinations with marked dissimilarities in image characteristics such as color hue, brightness and contrast, the difference in ulcer shape and size, and the quality of preparation. Another challenge in developing machine learning algorithms is the adaptation of algorithms to various platforms and capsule types (one-head capsule, two-head capsule, different manufacturers, and future capsule endoscopies).

The aim of our study was to create an accurate cross-domain model algorithm for the identification and grading of ulcers on CE, using two models of CE (PillCam Crohn and PillCam SB3, Medtronic) in CD patients.

## 2. Materials and Methods

### 2.1. Study Design

We randomly selected CE videos from patients diagnosed with CD as well as healthy subjects from our database and downloaded de-identified images from both ulcerated and normal mucosa. All patients were diagnosed and followed by the department of gastroenterology at Sheba Medical Center. The images were obtained by PillCam SB3 capsule and PillCam Crohn capsule (Medtronic Ltd., Dublin, Ireland) and reviewed with Rapid 9 (Medtronic Ltd., Dublin, Ireland) capsule reading software. The extracted images were labeled by gastroenterology fellows supervised by capsule experts. Both ulcers and erosions were considered “ulcerated mucosa.” For patients diagnosed with CD, we aimed to extract a comparable number of pathological and normal images. An institutional review board granted approval for this retrospective study.


**Identification of small bowel ulcerated mucosa (Experiments 1–3):**


Classification of CE images of the small bowel into normal mucosa and ulcerated mucosa was evaluated through 3 experiments:

Experiment 1: The model was trained on CE images from each of the capsules separately and was tested on images from the same capsule type.

Experiment 2 and experiment 3 evaluated “domain transfer”:

Experiment 2: The model that had been trained on CE images from each capsule type (in experiment 1) was tested on CE images from the other capsule type, i.e., the model was trained on CE images from the PillCam SB3 capsule and was tested on CE images from PillCam Crohn capsule, and vice versa (cross-domain).

Experiment 3: The model was trained and tested on a combined dataset of CE images from both capsule types (combined model).


**Identification of colonic ulcerated mucosa (Experiment 4):**


Classification of CE images of the colon into normal mucosa and ulcerated mucosa was done by training and testing the model on CE images from the PillCam Crohn capsule.


**Ulcer grading (Experiment 5):**


The model was trained and tested on a combined dataset of CE images of the small bowel from both capsule types. Differentiation between Grade 1 (mild) ulceration and Grade 3 (severe) ulceration was evaluated. Grade 2 (intermediate) ulcerations were omitted due to anticipated label noisiness. As they have intermediate properties, according to a previous study [28], the performance of machine learning (ML) algorithms is expected to be lower in their definition, and even the agreement among readers is lower.

Software and hardware:

The models were developed on Python (ver. 3.6.5, 64 bits) utilizing the open-source Pytorch and Pytorch Lightning libraries as the backend for CNN algorithms and the open-source Scikit-Learn library (ver. 0.20.2) for evaluation metrics algorithms. Models trained and evaluated on an Intel i7 CPU and Tesla V-100 GPU.

### 2.2. Neural Network Model

Deep learning is a subtype of AI mainly involving artificial neural networks. CNN, a subtype of deep learning, is optimized for solving computer vision tasks by employing pattern recognition [34]. EfficientNet is a CNN architecture and scaling method that uniformly scales all dimensions of depth/width/resolution using a compound coefficient. The compound scaling method is justified by the intuition that if the input image is bigger, then the network needs more layers to increase the receptive field and more channels to capture more fine-grained patterns on the bigger image. The base EfficientNet-B0 network is based on the inverted bottleneck residual blocks of MobileNetV2, in addition to squeeze-and-excitation blocks. We used EfficientNet-B4 as the training network. Google’s EfficientNet networks family has shown state-of-the-art results on the ImageNet dataset [35]. The network’s weights were initialized using weights from the 1.2 million ImageNet everyday color images. All the computer vision tasks in the study were binary classification tasks using binary cross-entropy loss. The models’ final neurons were sigmoid neurons for outputting class probabilities. The network was trained on capsule endoscopy images. In experiments 1 and 2, only a single type of capsule was used for training, while in the multi-domain case (experiment 3), both the PillCam SB3 capsule and the PillCam Crohn capsule images were used for training, thus letting the network learn from 2 different modalities. The preprocessing of capsule images included cropping of images’ borders and legends. Images were then resized into a 516 × 516 matrix, and pixels were normalized into 0–1 by dividing by 255.

The following parameters were used for training the network:-For ulcer identification: 3 epochs; batch size 12; Adam optimization with a learning rate of 5 × 10^−5^. The network output was a binary classification layer: non-ulcerated mucosa versus ulcerated mucosa images.-For ulcer grading: 3 epochs; batch size 12; Adam optimization with a learning rate of 5 × 10^−5^. The network output was a binary classification layer: grade 1 (mild) ulcerations vs. grade 3 (severe) ulcerations.

### 2.3. Class Activation Maps

Class activation maps [CAM] were used to analyze which image regions led to the network’s classification decisions of ulcers. For this purpose, we have applied gradient-weighted class activation mapping [Grad-CAM] [36]. This algorithm uses the gradients of the target label, flowing into the final convolutional layer, to produce a coarse localization map highlighting the important regions in the image.

### 2.4. Metrics

Accuracies were calculated using a cut-off probability of 0.5. Receiver operating curves [ROC] were plotted for the network results by varying the operating threshold. The area under the ROC curve [AUC] and accuracies were calculated both patient-wise and for each of the 5 folds. Sub-analyses included the AUCs of ulcerated mucosa vs. non-ulcerated mucosa images and grade 1 vs. grade 3 ulceration images.

## 3. Results

### 3.1. Study Population

The entire dataset included 33,100 CE images. There were 20,621 PillCam SB3 CE images and 12,479 PillCam Crohn CE images, of which 3582 are from the colon. Considering the findings, there were 15,684 normal mucosa images and 17,416 ulcerated mucosa images. Data regarding small bowel CE images are presented in Table 1.

For the colon, we collected PillCam SB3 CE images: 1597 normal mucosa images and 1985 ulcerated mucosa images. Colonic ulcerated mucosa images were not graded.

### 3.2. Identification of Small Bowel Ulcerated Mucosa (Experiments 1–3)

The dataset included 20,621 PillCam SB3 CE images and 12,479 PillCam Crohn CE images of the small bowel.


**Experiment 1: Separate models**


The model was trained and tested separately on CE images from each capsule type. In both capsule types, there was excellent accuracy in classification, with the area under the curve (AUC) over 0.95 in the majority. For each capsule type, over five different folds, with non-overlapping patients in each, ROC AUC, accuracy, and mean patient accuracy are presented in Table 2 and shown in Figure 1.


**Experiment 2: Cross Domain**


To justify the need for a global model, the models trained on each of the capsule types (“domains”) separately and were tested on the other capsule-type images. With the model trained on the PillCam SB3 CE images and tested on the PillCam Crohn CE images, the accuracy and mean patient accuracy were 0.569 and 0.545, respectively. However, it accomplished a high AUC of 0.921 (Figure 2a). With the model trained on the PillCam Crohn CE images and tested on the PillCam SB3 CE images, it achieved high accuracy (0.877) and a mean patient accuracy of 0.88, with an AUC of 0.948 (Figure 2b).


**Experiment 3: Combined model**


The model was trained and tested on a combined dataset of CE images from both capsule types and achieved excellent accuracy in the identification of ulcerated mucosa with an area under the curve (AUC) of over 0.98. Over five different folds, with non-overlapping patients in each, the ROC AUC, accuracy, and mean patient accuracy are presented in Table 3 and shown in Figure 3.

### 3.3. Identification of Colonic Ulcerated Mucosa (Experiment 4)

The model was trained and tested on 3582 CE images of PillCam Crohn capsule: 1597 Normal mucosa images and 1985 Ulcerated mucosa images. There was excellent accuracy in the identification of colonic ulcerated mucosa with an area under the curve (AUC) of over 0.98 in the majority. Over five different folds, with non-overlapping patients in each, the ROC AUC, accuracy, and mean patient accuracy are presented in Table 4 and shown in Figure 4.

### 3.4. Ulcer Grading (Experiment 5)

This experiment included 10,898 CE images of small bowel ulcerated mucosa from both capsule types. The model was pre-trained on both types as the combined model led to the best results in ulcer identification. There was excellent accuracy in classification ulcerations to grade 1 and grade 3 with an area under the curve (AUC) of 0.99. Over five different folds, with non-overlapping patients in each, the ROC AUC, accuracy, and mean patient accuracy are presented in Table 5, and ROC curves for each fold are shown in Figure 5.

### 3.5. Class Activation Map

Class activation maps [CAM] were used to analyze which image regions led to the network’s classification of images. The gradient-weighted class activation mapping [Grad-CAM] produces a coarse localization map highlighting the important regions in the image. Applying this algorithm to the images enables the visual presentation of the ulcer area in the CE images (Figure 6).

## 4. Discussion

In recent years several studies have examined the utility and efficacy of ML in the identification of different pathologies in small bowel mucosa. Previous studies [25,26,27,28,29,30,31,32,33] have demonstrated the high accuracy of ML models in the identification and grading of small bowel ulcers in CE images in CD patients. Recent metanalysis showed a combined sensitivity of 93% and a combined specificity of 92% [37] in the identification of gastrointestinal ulcers. These studies are almost exclusively limited to a single capsule type, meaning the ML model is trained and tested on the CE images of one capsule type.

Our study shows that the CNN algorithm was able to detect ulcerations in established CD patients with an AUC of 0.98 and above. Applying the same algorithm on CE images originating in both types of capsules, PillCam SB3 and PillCam Crohn capsules, produced an accurate diagnostic capability with high AUC.

The present study reinforces the results of previous studies on the ability of ML to detect and rank small bowel ulcers of CD patients in two types of capsules commonly used worldwide. In addition, it shows the same ability to detect ulcers in the colon. The uniqueness of this study is in the development of a combined model for two different capsules, which not only did not reduce its accuracy but also raised it. This is part of the refinement required in ML models on the way to adapting them to clinical practice, where we hope there will be one algorithm that can be adapted to all existing and future capsule types.

Recently, Houdeville et al. [38] addressed this issue in a different clinical setting, the detection of angiectasias in small bowel CE images. They evaluated an ML algorithm trained on CE images from PillCam SB3 capsule on CE images from another manufacturer, i.e., Mirocam’s capsule. The model achieved high sensitivity and specificity (96.1% and 97.8%, respectively). This resembles part of experiment 2 (cross-domain) in our study, but with different lesions and with only one direction, meaning they did not check training the model on Mirocam’s and testing it on PillCam SB3 CE images. In our study, the cross-domain model revealed high accuracy when trained on the PillCam Crohn CE images and tested on the PillCam SB3 CE images but with low accuracy (0.5) in the opposite situation. Their study has the advantage of dealing with different manufacturers’ capsules. However, in our study, we took another step and examined a combined model on a combined dataset of CE images from two capsule types, analyzing both identification of ulcers and grading ulcers.

Another aspect examined in our study is the analysis of the performance of the model on individual patients, i.e., the identification of ulcers in a set of images from the same patient. The individual patient-level analysis provided high and consistent diagnostic accuracy with shortened reading time. This is also more appropriate for clinical practice, where the algorithm is supposed to read one capsule of one patient at a time.

As for misdetections, false negative errors may be secondary to the small diameter of the ulcer or its suboptimal visibility due to contents in the bowel cavity. When it comes to the diagnosis of ulcers and aphthae in Crohn’s disease, the clinical importance of missing a single ulcer or aphthae is low and has significance only in the quantification of the inflammation. The importance of misdetections can be more significant in the diagnosis of other pathologies, such as angiodysplasias or polyps. Regarding visibility, future algorithms should include an assessment of the degree of cleanliness so that the DL will alert in case of low visibility.

This study had several limitations: first, the analysis was devoted to ulcerations and aphthae only; any other given pathology will require similar training. In the future, there might be a combined model that includes all the possible SB lesions. Second, although we used CE images from two different capsule types, they both are from the same manufacturer, suggesting they may have common imaging characteristics that contributed to the excellent accuracy of the combined model. Future studies should include images from different capsules from different manufacturers. Third, the number of evaluated colon images in this study was substantially smaller than the number of evaluated small bowel images. However, this is only a secondary outcome of this study since the colonic images are from one capsule type, i.e., PillCam SB3, and a combined model for colonic images is out of the scope of this study. Future studies should also focus on additional small bowel and colonic pathologies, including other inflammatory etiologies. Finally, as in previous studies, we also used separate still CE images in the analysis, but the next step should be to identify and quantify signs of inflammation in an individual film.

## 5. Conclusions

While single capsule models performed well on validation sets from the same domain, they performed poorly on the other capsule’s test sets. Developing a combined model for two different capsules provided high and consistent diagnostic accuracy. Creating a holistic AI model for automated capsule reading is an essential part of the refinement required in ML models on the way to adapting them to clinical practice.

## Figures and Tables

**Figure 1 diagnostics-12-02490-f001:**
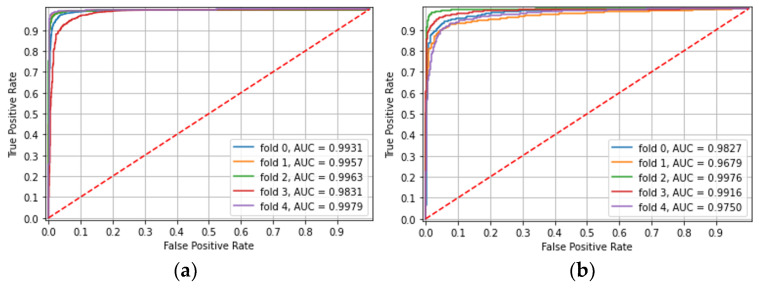
Experiment 1-Identification of small bowel ulcerated mucosa according to capsule type: (**a**) PillCam SB3 CE images, (**b**) PillCam Crohn CE images.

**Figure 2 diagnostics-12-02490-f002:**
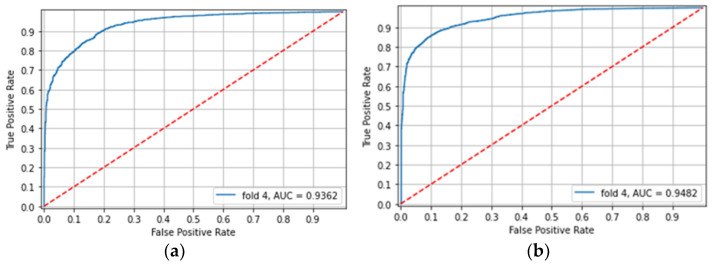
Experiment 2: (**a**) Classification of PillCam Crohn CE images by a model trained on PillCam SB3 CE images (**b**) Classification of PillCam SB3 CE images by a model trained on PillCam Crohn CE images.

**Figure 3 diagnostics-12-02490-f003:**
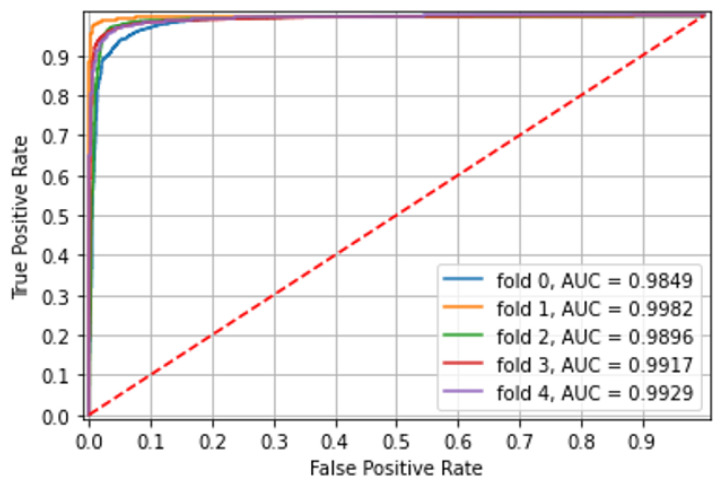
Experiment 3—Classification of PillCam Crohn CE images by a combined model.

**Figure 4 diagnostics-12-02490-f004:**
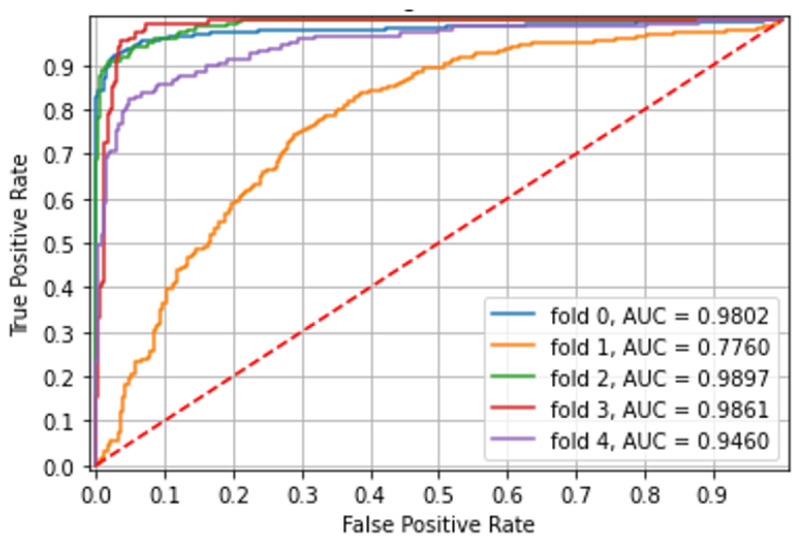
Experiment 4—Identification of colonic ulcerated mucosa.

**Figure 5 diagnostics-12-02490-f005:**
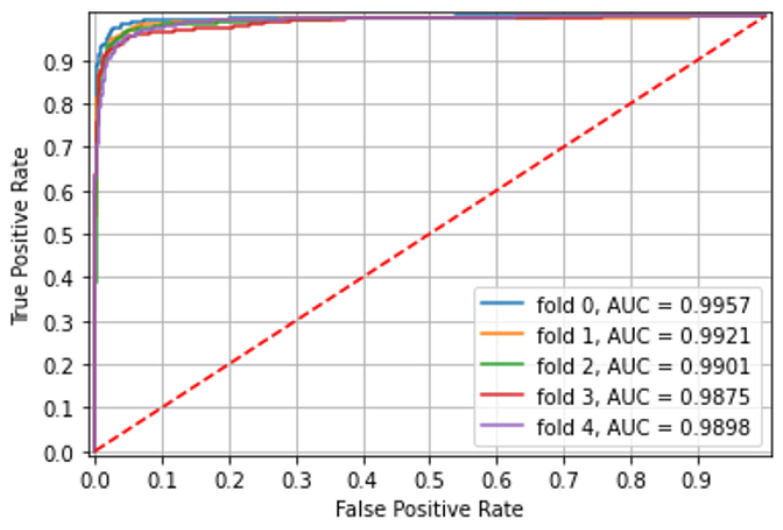
Experiment 5—Grading of small bowel CE mucosal ulcerations images.

**Figure 6 diagnostics-12-02490-f006:**
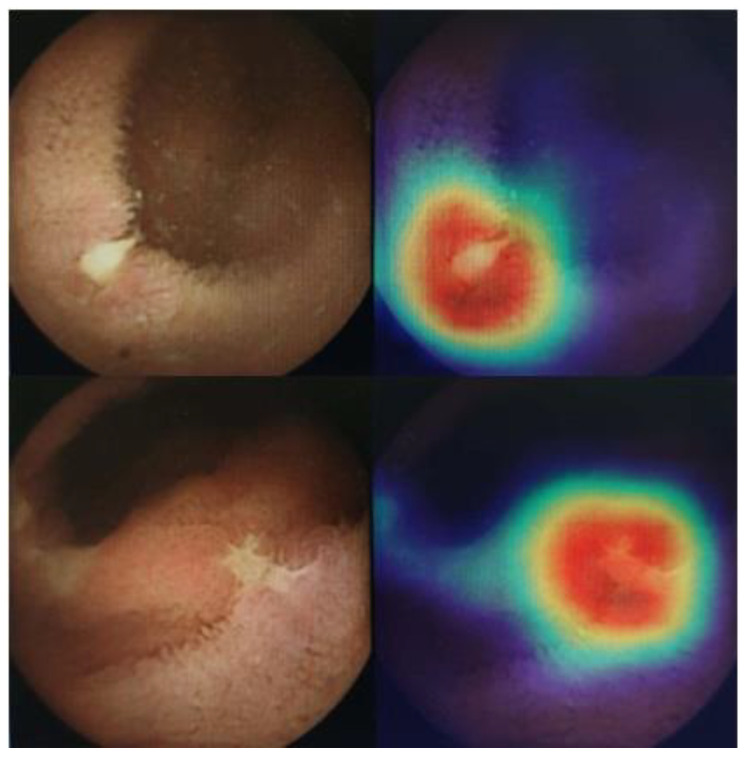
Class activation map.

**Table 1 diagnostics-12-02490-t001:** Small bowel CE images.

	PillCam SB3	PillCam Crohn	Total
Normal	10,248	3839	14,087
Ulcer grade 1	5398	2147	7545
Ulcer grade 2	3006	1527	4533
Ulcer grade 3	1969	1384	3353
Total	20,621	8897	29,518

**Table 2 diagnostics-12-02490-t002:** Experiment 1-Identification of small bowel ulcerated mucosa according to capsule type.

Pillcam SB3 CE Images				PillCam Crohn CE Images			
Fold	Accuracy	Mean Patient Accuracy	ROC_AUC	Fold	Accuracy	Mean Patient Accuracy	ROC_AUC
0	0.966	0.977	0.993	0	0.935	0.893	0.982
1	0.978	0.976	0.995	1	0.915	0.908	0.967
2	0.974	0.985	0.996	2	0.978	0.977	0.997
3	0.935	0.98	0.983	3	0.954	0.954	0.991
4	0.975	0.982	0.997	4	0.924	0.918	0.974

**Table 3 diagnostics-12-02490-t003:** Classification of PillCam Crohn CE images by a combined model.

Fold	Accuracy	Mean Patient Accuracy	ROC_AUC
0	0.941	0.978	0.984
1	0.975	0.975	0.998
2	0.958	0.982	0.989
3	0.963	0.967	0.991
4	0.963	0.968	0.992

**Table 4 diagnostics-12-02490-t004:** Experiment 4—Identification of colonic ulcerated mucosa.

Fold	Accuracy	Mean Patient Accuracy	ROC_AUC
0	0.896	0.951	0.98
1	0.478	0.74	0.77
2	0.937	0.948	0.989
3	0.931	0.905	0.986
4	0.828	0.795	0.946

**Table 5 diagnostics-12-02490-t005:** Experiment 5—Grading of small bowel CE mucosal ulcerations images.

Fold	Accuracy	Mean Patient Accuracy	ROC_AUC
0	0.972	0.979	0.995
1	0.965	0.954	0.992
2	0.960	0.925	0.990
3	0.950	0.916	0.950
4	0.948	0.932	0.989

## Data Availability

Data are available by correspondence to the Corresponding Author.

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
