# Peer review of "Deep Learning Multi-Domain Model Provides Accurate Detection and Grading of Mucosal Ulcers in Different Capsule Endoscopy Types"

_diagnostics, 2022, doi:10.3390/diagnostics12102490_

Round 1

Reviewer 1 Report

This is an interesting study evaluating the role of artificial intelligence algorithms in the identification and grading of ulcers in capsule endoscopy images. The study design is good and the results are well presented and important. I only have some minor comments:

1.  The number of evaluated colon images in this study was substantially smaller than the number of evaluated small bowel images, and this should be mentioned as a possible study limitation

2.      2.  The study evaluated the model performance in the identification of colonic ulcers using the Pillcam Crohn capsule and it would be interesting to also assess its role in the Pillcam second generation colon capsule. As the authors mentioned, capsules from other manufacturers were not included and these may be listed as potential areas for future research.

3.      3.  I wonder what the effect of bowel prep quality may be in the identification of ulcers, particularly in the colon. Did the authors select only images with good bowel prep quality, or also images with suboptimal/poor bowel prep?

4.      4. The evaluated models were used to identify ulcerations compared to normal mucosa. It would be interesting to know if they can also differentiate ulcers from other small or large bowel findings, such as cystic lymphangiectasias or polyps etc. This would really increase the clinical utility of such models, and I guess should be the focus of future research. I would suggest the authors add some comments in their discussion.

     Overall, it is a nice paper on an important topic and the manuscript is well presented.

Author Response

We appreciate the reviewer’s insightful and helpful comments on our manuscript. 

Please see our detailed response below:

  1. The number of evaluated colon images in this study was substantially smaller than the number of evaluated small bowel images, and this should be mentioned as a possible study limitation Thank you for the comment. Indeed the number of colonic images was lower d/t the fact that we used SBIII capsule (that only obtained small bowel images) and Pillcam Crohn's (both small bowel and colon). We addressed this limitation in the discussion.
  2.  The study evaluated the model performance in the identification of colonic ulcers using the Pillcam Crohn capsule and it would be interesting to also assess its role in the Pillcam second generation colon capsule. As the authors mentioned, capsules from other manufacturers were not included and these may be listed as potential areas for future research. Indeed it can be interesting to assess the the performance in the Pillcam second generation colon capsule however this was out of scope of our research. Incorporation of additional capsule models is our research priority as well.
  3. I wonder what the effect of bowel prep quality may be in the identification of ulcers, particularly in the colon. Did the authors select only images with good bowel prep quality, or also images with suboptimal/poor bowel prep? We selected only images with adequate visibility and quality as only those are accurate and diagnostic to the human reader.
  4. The evaluated models were used to identify ulcerations compared to normal mucosa. It would beinteresting to know if they can also differentiate ulcers from other small or large bowel findings, such as cystic lymphangiectasias or polyps etc. This would really increase the clinical utility of such models, and I guess should be the focus of future research. I would suggest the authors add some comments in their discussion. This study is devoted to ulcerations and aphthae only, and this is one of the limitations. This is another step in developing the optimal DL model in the future that be able to read different capsule models and recognize different lesions. In future studies it will also be very useful to focus on additional etiologies of inflammatory lesions.

Sincerely,

Dr Reuma Margalit Yehuda

Reviewer 2 Report

In this paper, the authors presented the deep learning multi-domain model to detect and grade the mucosal ulcers by different capsule endoscopes. The data set mainly includes 33100 capsule endoscope pictures, including 20621 Pillcam-sb3 pictures and 12479 PillCam-Crohn pictures. There are totally five experiments. In the first experiment, the data of model training and testing are from the same capsule types; in the second experiment, the data of model training and testing are from different capsule types; in the third experiment, the data of model training and testing are from the combination of two capsule types. The fourth experiment is about the recognition of colonic ulcers, and the fifth experiment is the recognition of the mucosal ulcer grades.

The topic of this paper is significant. The paper is referrable to the colleagues in this area. Also, the paper organizations and the English usage are acceptable. Suggestions for the paper:

1.     Give more technical information about the deep learning multi-domain model.

2.     For the experimental results, such as the detection accuracy, it is better to give more discussions about the reasons for the mis-detections.

3.     The conclusion of the presented work is that the model trained by combining two different capsule pictures has higher and more consistent diagnostic accuracy, but this paper is only aimed at the recognition accuracy of mucosal ulcers. The lesions of the small intestine, including bleeding, polyps and other symptoms, are not involved in their experiments. It is suggested that the future follow-up experiments should include the recognition of other lesions of small intestine.

Author Response

We appreciate the reviewer’s insightful and helpful comments on our manuscript. 

Please see our detailed response below:

Suggestions for the paper:

  1. Give more technical information about the deep learning multi-domain model. We agree and we added more information.
  2. For the experimental results, such as the detection accuracy, it is better to give more discussions about the reasons for the mis-detections. Thank you for the comment, we added it in the discussion.
  3. The conclusion of the presented work is that the model trained by combining two different capsule pictures has higher and more consistent diagnostic accuracy, but this paper is only aimed at the recognition accuracy of mucosal ulcers. The lesions of the small intestine, including bleeding, polyps and other symptoms, are not involved in their experiments. It is suggested that the future follow-up experiments should include the recognition of other lesions of small intestine. We agree with the reviewers' comment. We are planning to expand our future experiments to include additional small bowel pathologies.

Sincerely,

Dr Reuma Margalit Yehuda